# Diagnosis and Diagnostic Challenges of Secondary Mitral Regurgitation in the Era of Transcatheter Edge-to-Edge Repair of the Mitral Valve

**DOI:** 10.3390/jcm14134518

**Published:** 2025-06-26

**Authors:** Yusef B. Saeed, Kyra Deep, Andreas Hagendorff, Bhupendar Tayal

**Affiliations:** 1Cleveland Medical Center, University Hospitals, Cleveland, OH 44106, USA; yusef.saeed@uhhospitals.org (Y.B.S.); kyra.deep@uhhospitals.org (K.D.); 2Department of Cardiology, Leipzig University Hospital, 04103 Leipzig, Germany; andreas.hagendorff@medizin.uni-leipzig.de; 3Division of Cardiology, University of Arkansas for Medical Sciences, 4301 W Markham Street, Little Rock, AR 72223, USA

**Keywords:** secondary mitral regurgitation, ventricular functional mitral regurgitation, atrial functional mitral regurgitation, transcatheter edge-to-edge repair of the mitral valve, heart failure, MitraClip, echocardiography, semiquantitative methods, quantitative methods, cardiac magnetic resonance imaging

## Abstract

Secondary mitral regurgitation (sMR) is commonly understood to be secondary to heart failure (HF), left ventricular (LV) dilation, and altered coaptation of the mitral annulus. Three forms of sMR exist: non-ischemic sMR, ischemic sMR, and atrial functional sMR. In the past, there have been limited treatment options for this condition besides medication. Recently, the management of sMR has been revolutionized by the recent advances in percutaneous transcatheter edge-to-edge repair of the mitral valve (m-TEER). However, the major trials investigating this technology have shown that appropriate patient selection is of critical importance to achieve benefit. As such, there is a renewed interest in the accurate diagnosis of sMR. Herein, we review the etiology, management, and diagnosis of sMR in the era m-TEER.

## 1. Introduction

This review will focus on the etiology, management, and diagnosis of sMR. As opposed to primary MR, which is a disease of the mitral valve (MV) apparatus, sMR is considered a distinct clinical entity with a worse prognosis and is a disease of the ventricle or atria leading to left atrial dilation and enlargement of the mitral annulus [1,2,3]. Traditionally, sMR is thought to be related to deleterious cardiac chamber remodeling, which affects a structurally normal valve leading to insufficient coaptation, after which a vicious cycle ensues wherein increasing MR begets further LV dilation, and thus further MR [2,4,5].

## 2. Etiology of Secondary MR

### 2.1. Non-Ischemic Secondary MR

The three major forms of sMR are summarized in Figure 1. Underlying dilated cardiomyopathy and decreased LV closing forces can lead to distortion of the LV geometry tethering the structurally normal mitral leaflets from papillary muscle displacement, leaflet tethering, or annular dilation, which can result in inadequate leaflet coaptation, and, consequently, non-ischemic sMR [6]. Reduced closing forces from reduced contractility, electrical or mechanical dyssynchrony of papillary muscles or of intra-left ventricular conduction system may also contribute [7]. Annular dilation and loss of annular contraction are thought to be the predominant mechanism in non-ischemic sMR, which leads to a central jet of MR [8].

### 2.2. Ischemic Secondary MR

Ischemic sMR typically is secondary to chronic ischemia, resulting in post-infarction LV remodeling which has a similar mechanism as mentioned above but involves regional wall motion abnormalities and tethering of the posterior leaflet, often leading to a posteriorly directed jet of MR [4]. A central jet of MR can also be seen if there is global LV dysfunction related to multivessel ischemic disease [8]. Acute MR is rare and occurs after papillary muscle rupture or severe ischemia in the left circumflex and/or right coronary artery [9].

### 2.3. Atrial Functional MR

Atrial functional MR is commonly seen in patients with atrial dysfunction (from chronic atrial fibrillation or myopathy) and heart failure with a preserved ejection fraction (HFpEF) which leads to a dilated left atrium and, consequently, annular dilation and insufficient leaflets adaptation with macroscopically normal leaflets [10]. However, unlike the other mechanisms, the LV geometry and function is preserved [11,12]. Whereas leaflet motion in ischemic sMR is typically restricted, classified as Carpentier Type IIIb, leaflet motion in atrial functional MR and non-ischemic MR is typically normal, classified as Carpentier Type I [8,9,10]. An exception to this is ‘atriogenic hamstringing’, which is a form of atrial functional MR that resembles Carpentier Type IIIb leaflet motion due to posterior displacement of the mitral annulus from an enlarged LA, leading to the tethering of the posterior mitral leaflet [13,14,15]. LV ejection fraction (EF) is preserved in this form of sMR in contrast to other types of sMR.

## 3. Management of Secondary MR with Transcatheter Edge-to-Edge Repair (TEER) of Mitral Valve

Whereas isolated surgical intervention has been indicated for primary MR, it has not been strongly indicated for sMR [1,2,16]. Guideline-directed medical therapy (GDMT) and cardiac resynchronization therapy (CRT) are the mainstays of treatment in this population. As medical therapy is maximized and volume status and loading conditions improve, the severity of disease is likely to change. The need for intervention on MR is often evaluated only after optimizing GDMT, specifically the maximum tolerated dose of a beta-blocker and ACE inhibitor [17]. Therefore, complete imaging and assessment of MR should be performed after medical therapy is optimized, as this could change both the patient’s symptoms and technical severity of MR.

Certain cases of HF and MR are refractory to GDMT and CRT [3,7,18,19,20,21]. M-TEER is an emerging treatment for mitral regurgitation which reapproximates the A2 and P2 scallop of the MV leading to coaptation. Over the years, multiple trials with m-TEER devices have demonstrated a significant reduction in HF hospitalization [22,23,24] and cardiovascular mortality [22,23,25], as well as improvement in functional status and MR severity [24,26,27,28,29,30]. Initial trials utilized the Abbot MitraClip system [31,32], whereas more recent trials have utilized the Edwards PASCAL system [14,29,33,34]. Table 1 presents a summary of completed and ongoing trials for sMR. Endovascular Valve Edge-to-Edge Repair Study (EVERST II) included primary MR patients but is included as it was the first major randomized controlled trial (RCT) [35,36]. These trials were performed across different countries over almost two decades with a heterogeneous population with consistent findings of benefit across the spectrum of HF patients with significant sMR. Recent retrospective studies have also clarified the benefit of m-TEER in atrial functional MR [37,38,39]. The Cardiovascular Outcomes Assessment of the MitraClip Percutaneous Therapy for Heart Failure Patients with Functional Mitral Regurgitation Post Approval Study (COAPT PAS) registry is a multicenter prospective study which has included the most patients to date and has shown procedural success and efficacy of M-TEER in various sMR phenotypes [30]. There have been several RCTs which have led to a paradigm shift in sMR management with mTEER [22,23,35,40,41,42,43,44], some which have conflicting findings [45], and one which has demonstrated noninferiority and superior safety profile of m-TEER as compared to MV surgery [44]. Several other percutaneous therapies for sMR are also listed in Table 1 including transcatheter mitral loop cerclage, mitral valve replacement, and annuloplasty [46,47,48,49,50,51,52,53,54,55,56,57,58].

### 3.1. Pivotal RCTs That Transformed sMR Management with M-TEER

There have been three major RCTs that compared M-TEER and GDMT to GDMT alone in the patients with sMR. The Cardiovascular Outcomes Assessment of the MitraClip Percutaneous Therapy for Heart Failure Patients with Functional Mitral Regurgitation (COAPT) trial demonstrated that the addition of m-TEER leads to a significantly reduced HF hospitalization rate, all-cause mortality, and cardiovascular mortality at the 5-year follow-up [22,25]. The results of the COAPT trial were notably contrasted with the findings of the Percutaneous Repair with the MitraClip Device for Severe Functional/Secondary Mitral Regurgitation (MITRA-FR) trial. The MITRA-FR Trial investigated m-TEER in a similar patient population with apparent severe MR but did not show that m-TEER had a significant impact on mortality or HF hospitalizations as compared to GDMT alone [42]. Six years later, the Randomized Study of the MitraClip Device in Heart Failure Patients With Clinically Significant Functional Mitral Regurgitation (RESHAPE-HF2) trial showed that, at 2-year follow-up, m-TEER was associated with reduced hospitalization, increased quality of life, but no difference in mortality [43]. Interestingly, these three RCTs studied patients with similar pathology but the results of COAPT and RESHAPE-HF2 indicated remarkable efficacy and safety of m-TEER whereas MITRA-FR did not even show symptomatic or functional improvement [45,59]. It is also notable that COAPT demonstrated cardiovascular mortality and HF hospitalization benefit with M-TEER, whereas RESHAPE-HF2 only found HF hospitalization benefit [22,42,43,60]. Recently, a study level meta-analysis of these three RCTs was showed lower HF hospitalizations and improved functional capacity, but it did not show improvement in all-cause or cardiovascular mortality [61].

There are two other relevant RCTs that are important to mention. The Transcatheter Mitral Valve Repair in non-Responders to Cardiac Resynchronization Therapy (MITRA CRT) was a smaller RCT that investigated response to M-TEER in patients with dilated cardiomyopathy who were ‘nonresponders’ to GDMT and CRT (mean effective regurgitant orifice area [EROA] 54 ± 46 mm^2^). Despite the larger EROA, M-TEER was associated with a lower composite endpoint of cardiovascular death, heart transplantation, and HF hospitalization than the control group at 12 months (13% vs. 57%, respectively, *p* value 0.003) [23]. Lastly, The Multicenter Mitral Valve Reconstruction for Advanced Insufficiency of Functional or Ischemic Origin (MATTERHORN) trial expanded on the work of the above RCTs in that it compared m-TEER to surgical MV replacement in a study population with less advanced cardiomyopathy (EROA 20 ± 10 mm^2^, LV end diastolic volume [EDV] 164.6 ± 57.3 mL) [44,62]. MATTERHORN advanced the role of m-TEER as it was found to be noninferior in regard to efficacy and significantly safer than MV surgery at 1 year.

### 3.2. Conceptualizing the Diverging Results of MITRA-FR with the Other Trials

The trial design, baseline characteristics, and outcomes of the RCTs mentioned above are listed in Table 2. As far as inclusion criteria, all three trials included moderate to severe MR but COAPT and RESHAPE-HF2 used guidelines that defined it at a lower threshold as compared to MITRA-FR [63,64,65]. Moreover, COAPT specifically excluded patients with non-ambulatory New York Heart Association Stage IV and LVESD > 70 mm in contrast to MITRA-FR and RESHAPE-HF2 [45,59,65,66,67,68]. Importantly, GDMT optimization was more rigorously performed pre-trial in COAPT and RESHAPE-HF2 as compared to MITRA-FR which utilized a more ‘real world’ approach to GDMT adjustment [65,67]. RESHAPE-HF2 had a higher proportion of patients on quadruple medical therapy, especially with ARNI, MRA, and SGLT-2 inhibitors, albeit SGLT-2 Inhibitor utilization was low [24,62]. Because of these reasons, it is widely thought that MITRA-FR’s study population had comparatively more advanced cardiomyopathy and worse medical optimization [24,68].

In addition to the reasons above, the discordant results of MITRA-FR with the other abovementioned RCTs have been further conceptualized utilizing proportionate versus disproportionate hypothesis of sMR. This theory posits that sMR that is disproportionate to LV size or indirectly to the existing cardiomyopathy (lower EROA to LV EDV ratio) and is thought to be more amenable to MV intervention, whereas sMR proportionate to LV dilation (higher EROA to LV EDV ratio) is thought to be more amenable with pharmacologic therapy (Figure 2) [69]. The inclusion of patients with higher EROA and lower LV size in COAPT is hypothesized to represent disproportionate sMR caused by valvular dysfunction, as opposed to advanced LV disease and dysfunction, which is readily correctable by m-TEER. In contrast, MITRA-FR’s lower EROA and higher LV size is thought to represent proportionate sMR which is less likely to benefit m-TEER and probably more amenable to pharmacologic therapy, with the trial’s lack of response to M-TEER further supporting the theory (Figure 3) [70]. However, RESHAPE-HF2 study population demonstrated the lowest mean EROA value, but still benefited from m-TEER. Taken together, the results of the three respective RCTs are in part explained by the severity of their underlying cardiomyopathy, with the RESHAPE-HF2 study population being early stage deriving some benefit, the MITRA-FR study population being late stage deriving little to no benefit, and the COAPT study population being in the ‘sweet spot,’ deriving the most benefit [14,71].

Although the proportionate versus disproportionate sMR theory is widely cited as an explanation for the above differences, retrospective studies have mixed support. The findings of retrospective analysis of the EuroSMR registry appeared to support this hypothesis as it stratified patients by EROA/LVEDV ratio into MR dominant (correlating with disproportionate sMR), MR-LV co-dominant, and LV dominant groups (correlating with proportionate sMR), and found increased mortality after m-TEER noted in the LV dominant group [72]. Similarly, a sub-study of the COAPT which showed that patients that resembled a MITRA-FR ‘proportionate’ sMR phenotype with EROA < 30 mm^2^ and LVEDVI > 96 mL/mm^2^ did not derive any additional benefit with m-TEER after GDMT [66,73]. However, the reciprocal situation was studied in patients with a COAPT-like ‘disproportionate’ sMR phenotype (ERO/LVEDV ratio > 0.15) in a post hoc analysis of MITRA-FR, along with numerous other echocardiographic subsets, none of which found any benefit with M-TEER [74]. Moreover, a recent metanalysis of proportional versus disproportional MR showed that there was no association of MR proportionality (EROA:LVEDV) to all-cause mortality or HF hospitalizations [75]. Alternative parameters that may have contributed to the findings of the above RCTs that have not yet been analyzed include RV function, LV fibrosis, and LV contractile reserve [75,76].

### 3.3. Guideline Recommendations Regarding sMR Severity and M-TEER Candidacy

MV intervention, whether surgical or percutaneous is typically offered when sMR is severe. Prior guidelines defined sMR severity with a lower cutoff, EROA ≥ 20 mm^2^ or regurgitant volume (RVol) as ≥ 30 mL, since MR severity can be underestimated in the presence of an elliptical orifice and in the setting of low stroke volume [77,78,79]. Although EROA ≥ 20 mm^2^ has been associated with increased mortality in sMR, this finding may be attributed to multi-morbid conditions and LV dysfunction. Surgical intervention in patients with moderate ischemic MR (EROA 0.20–39 mm^2^) was not associated with improved outcome, but rather had increased hazard of neurologic events and supraventricular arrhythmias [80,81]. Similarly, MITRA-FR showed no benefit for m-TEER in patients with a lower EROA. For these reasons, the lower cutoff for severe MR was thought to expose patients to unnecessary risk [81]. Accordingly, the most recent ACC/ASE, ACC/AHA and ESC/EACTS Guidelines define severe sMR as an EROA ≥ 40 mm^2^, RVol ≥ 60 mL, or regurgitant fraction (RF) ≥ 50% [2,81,82]. In the presence of an elliptical orifice, the ACC/ASE guidelines define severity by EROA ≥ 30 mm^2^ or RVol as ≥ 45 mL, or RF ≥ 40% [81]. The ESC/EACTS similarly defines severity by EROA ≥ 30 mm^2^ in the presence of elliptical orifice and, in low flow conditions, defines RVol as ≥ 45 mL [82]. The most recent ACC/AHA guidelines listed m-TEER as a Class 2a recommendation for severe sMR (defined as above) if HF symptoms persist despite GDMT initiation and supervision by a HF specialist [2]. Furthermore, candidacy for m-TEER was limited to patients with suitable mitral anatomy, LVEF < 50%, LVESD ≤ 70 mm, and pulmonary artery systolic pressure (PASP) ≤ 70 mm Hg, which more closely resembles the criteria utilized by the COAPT trial. The most recent ESC/EACTS similarly designated m-TEER as a Class 2a recommendation [82]. Importantly, these guideline-based eligibility criteria do not reflect the findings of RESHAPE-HF2 or MATTERHORN, both of which represent benefit of m-TEER in less severe MR (compared to prior studies, both studies had lower mean EROA and MATTERHORN had a higher mean EF) [14,31].

## 4. Diagnostic Challenges of Heart Failure and Secondary Mitral Regurgitation in the Era of Transcatheter Edge-to-Edge Repair of the Mitral Valve

The advent of m-TEER has fueled interest in properly defining severity of sMR as patient selection is key to achieving clinical benefit with this technology [79]. Diagnosing sMR is more of a challenge than primary MR as the structural remodeling in HF patients with sMR may be related to advanced cardiomyopathy rather than the MR itself, with significant contributions from LA/LV changes, advanced cardiomyopathy, and scar tissue [13,83]. Additionally, the LV is often dilated but hyperdynamic in primary MR leading to higher RVol which can fulfill the guideline-defined cutoffs for severity whereas the LV EF is lower in sMR with lower overall stroke volumes making it less likely for RVol to reach severity cutoffs [84]. Lastly, sMR is more dynamic than primary MR and may vary based on volume status, blood pressure, and the presence of vasoactive agents, complicating MR assessment [80,81]. Ultimately, a dedicated protocol for identifying patient with significant sMR diagnosis is essential.

### 4.1. Defining Secondary MR

Currently, three guidelines exist to aid in diagnosing sMR in MR including those from the ACC/ASE, ACC/AHA, and ESC/EACTs, and all recommend an integrative approach with utilizes both semiquantitative and quantitate methods [2,81,82]. Transthoracic echocardiography (TTE) is the first-line imaging modality due to its convenience [85]. Several qualitative and semiquantitative methods for quantifying MR have been delineated by the ASE and are reproduced in Figure 4a [81]. These include color flow Doppler, continuous wave Doppler (CWD), LA/LV volume and size, proximal isovelocity surface area (PISA), mitral inflow velocity, and vena contracta width (VCW). If these parameters are unanimously unremarkable then mild MR can be diagnosed, and vice versa. If only some of these parameters are present, quantitative methods can be utilized and/or further imaging with transesophageal echocardiography and cardiac MRI (CMR) can be pursued.

Despite the utility of the ASE algorithm (Figure 4a), we believe it may not be applicable to sMR as several of the qualitative and semiquantitative variables are affected by HF, independent of MR. We propose an algorithm specific to diagnosing sMR in Figure 4b. In our opinion, LA/LV size and volume and mitral inflow velocity are often unreliable in HF patients due to existing volume overload and variability with loading conditions. Color flow Doppler and CWD provide helpful visual estimates for MR severity, however they both may underestimate the MR severity in the presence of an eccentric jet. CWD showing a dense and complete jet suggests significant MR [81]. VCW is a surrogate marker for regurgitant orifice size with size ≥ 0.7 cm considered severe and size ≥ 0.4 cm considered moderate, which has been associated with higher mortality [80,86]. However, the threshold of 0.7 cm may need a revision as it may be too high to achieve in patients with sMR. Pulmonary vein systolic flow reversal (PVSFR) can be a very specific marker of significant sMR as it is recommended by ESC/EACTS [82]; however, a recent retrospective analysis of the COAPT trial found that presence of PVSFR was not an independent predictor of 2-year prognosis after m-TEER [87]. The presence or absence of PISA is also useful. Recently, RV dysfunction and RV-PA uncoupling have been associated with worse prognosis following M-TEER [88,89,90]. We feel that PVSFR and CWD (showing a dense, triangular, contoured wave) have the most utility in identifying sMR that is greater than mild in severity, assuming the regurgitant jet is not eccentric. If all the aforementioned parameters are present, this is sufficient for a diagnosis of severe sMR.

If all are absent except PVSFR or if there is ambiguity, we recommend further assessment with quantitative volumetric methods utilizing volume tracing or pulsed Doppler to measure RVol and RF or PISA-derived EROA calculation (Figure 4b). Importantly, volume tracing and pulsed Doppler rely on the continuity principle and are valid in the case of multiple or eccentric jets (in contrast to PISA-derived EROA) but are limited by nonlaminar flow, improper placement of sample volume, or atrial fibrillation which is common in HF [91].

Notably, the PISA-derived EROA method is the most commonly used approach for calculating sMR as it has been validated against LV angiography, has been shown to be prognostically meaningful after ischemic MR and can predict treatment response to m-TEER [22,42,92,93,94]. Accordingly, we feel that EROA ≥ 30 mm^2^ predicts a higher likelihood of response to therapy [22,42,43]. Importantly, a retrospective study of 177 patients with both ischemic and non-ischemic sMR showed superiority of the quantitative volumetric method over the PISA method in predicting mortality or heart transplant at a mean follow up of 3.7 years [95]. These findings may be attributed to the fact that the PISA method underestimates MR severity in the presence of noncircular jet orifice areas, eccentric jets, or nonholosystolic MR [80,85,96], or to its significant interobserver variability [80,81,97]. For these reasons, we prefer volume-tracing-derived RVol and RF for quantitation and utilize contrast to prevent underestimation of LV volumes or foreshortening of the LV apex [98]. Notably, calculation of both RVol and RF values is prudent as RVol ≥ 60 mL may not be observed due to underlying cardiomyopathy and low stroke volumes in sMR whereas RF may reach 40–50% [84]. Essentially, the RF differentiates between disproportionate versus proportionate MR as it is a reflection of RVol over total LV volumes. To this point, an observational study showed that outcomes of patients with intermediate severity sMR (EROA 20–29 mm^2^ and RVol as 30–44 mL) and RF ≥ 50% have been noticeably worse outcomes as compared to those with RF < 50% [79]. Some studies describe the incorporation of chest wall shape assessment using the modified Haller Index (MHI) to assess cardiovascular risk in patients with MR [99]. Given the simplicity of the MHI, it would be beneficial for future studies to explore its use in evaluating patients with sMR. For patients with multivalvular disease, intervention on a single valve will likely decrease the severity of disease on other valves. This has been demonstrated in sMR patients with concomitant aortic stenosis in multiple studies. Some have demonstrated the greatest improvement in sMR when there is not patient–prosthesis mismatch of the AVR [100,101,102]. This data suggests intervention on a diseased aortic valve should be performed prior to or in conjunction with intervention on a diseased mitral valve.

Overall, since accurate assessment of sMR informs candidacy for m-TEER and other MV interventions, a multiparametric, integrative approach should be utilized [103]. If 2-dimensional (2D) echocardiographic images are poor or nondiagnostic, we prefer CMR in these cases to accurately quantitate the MR. Transesophageal echocardiography (TEE) may offer higher resolution, but it is more commonly used for preprocedural and intraprocedural purposes as it provides detailed views of the MV apparatus [81].

Three-dimensional (3D) echocardiography may improve accuracy as compared to 2D as it can better quantify EROA and the vena contracta area. Three-dimensional echocardiography quantification of eccentric jets and complex MR make this imaging modality superior to 2D, which relies on geometric assumptions that commonly underestimate MR severity [94,104,105]. Three-dimensional imaging more frequently aligns with cardiac MR findings further proving its superiority to 2D echocardiography, but is ultimately limited by imaging complexity and expertise [106].

### 4.2. Emerging Role of Cardiac Magnetic Resonance Imaging

CMR is increasingly regarded as the gold standard for morphologic, volumetric, flow, and viability assessment and has been shown to have increased accuracy than 2D volumetric methods [97,107,108]. The two main techniques for MR quantification using CMR are the volumetric method and the phase-contrast method as shown in Figure 5 [109,110]. It has been demonstrated that EROA ≥ 40 mm^2^ on echocardiography was unable to predict indication for MV surgery as compared to CMR quantified RVol > 55mL in patients with moderate to severe primary MR [111]. As compared to echocardiographic parameters, RVol quantified by CMR is associated with prognostic significance regarding mortality, indication for MV surgery, and LV reverse remodeling after MV repair [107,108,109,110,111,112,113]. CMR also accurately and reliably quantifies LA volume and function and aids in identifying the etiology of cardiomyopathy and MR [97,108,109,114]. Furthermore, CMR plays a significant role in the assessment of myocardial scar and papillary muscle infarct, which carry prognostic information [110,115,116,117,118]. The presence of scars in both ischemic cardiomyopathy (ICM) and non-ischemic cardiomyopathy (NICM) significantly worsened outcomes in patients with sMR and RF ≥ 30% as compared to < 30% [106]. An observational study associated significantly worse outcome in patients with ICM with RF ≥ 35% and late gadolinium enhancement ≥ 5%, and in patients with NICM with RF ≥ 35% and late gadolinium enhancement ≥ 2%. All subgroups of ICM were found to have worsened outcomes as compared to NICM [116]. Lastly, MR severity on CMR has been defined as RVol ≥ 60 mL and RF ≥ 50% but the severity thresholds are not well established and may need revision [8,81,119].

### 4.3. Future Role of Artificial Intelligence

Machine learning has been utilized to integrate numerous clinical and imaging variables into distinct sMR phenotypes in order to prognosticate and predict treatment response [83,120]. In a cohort of HFrEF sMR patients on GDMT (n = 382), 32 morphologic and functional parameters of LV, LA, and RV were subjected to unsupervised machine learning principal component analysis, revealing four clusters of sMR patients [83]. The authors suggested that cluster 3 and 4 correlated with disproportionate and proportionate sMR, providing support to this theory [83]. The MITRA-AI study similarly utilized machine learning in a group of patients who underwent m-TEER (n = 822) and identified four clusters which each had differing mortality and hospitalization rates following m-TEER [121]. Lastly, the European Registry of Transcatheter Repair of Secondary Mitral Regurgitation (EuroSMR) registry has developed an artificial-intelligence-based risk score that utilizes 18 clinical, echocardiographic, laboratory, and medication parameters to predict a 1-year outcome following m-TEER that has outperformed existing scores [122]. In summary, artificial intelligence is allowing for a deeper understanding of m-TEER phenotypes, thereby improving risk stratification and patient selection.

## 5. Conclusions

In the past decade, there has been a significant advancement in the management of sMR with the advent of m-TEER. An accurate grading of sMR is of critical importance for the consideration of these novel percutaneous interventions. However, it is challenging to accurately quantify sMR and it may be underestimated in the presence of low flow status due to HF. Standard echocardiographic guidelines to quantify MR is widely available but may have to be tailored to apply them in the setting of HF. CMR can be a very useful tool in selected cases where sMR is ambiguous. Moreover, CMR adds prognostic information in patients with sMR, including etiologic, morphologic, volumetric, and viability assessment. Referral to high-volume centers is advisable when sMR presentation and quantification are equivocal, as these institutions possess specialized expertise in navigating the subtle complexities inherent in the diagnosis and management of this condition.

## Figures and Tables

**Figure 1 jcm-14-04518-f001:**
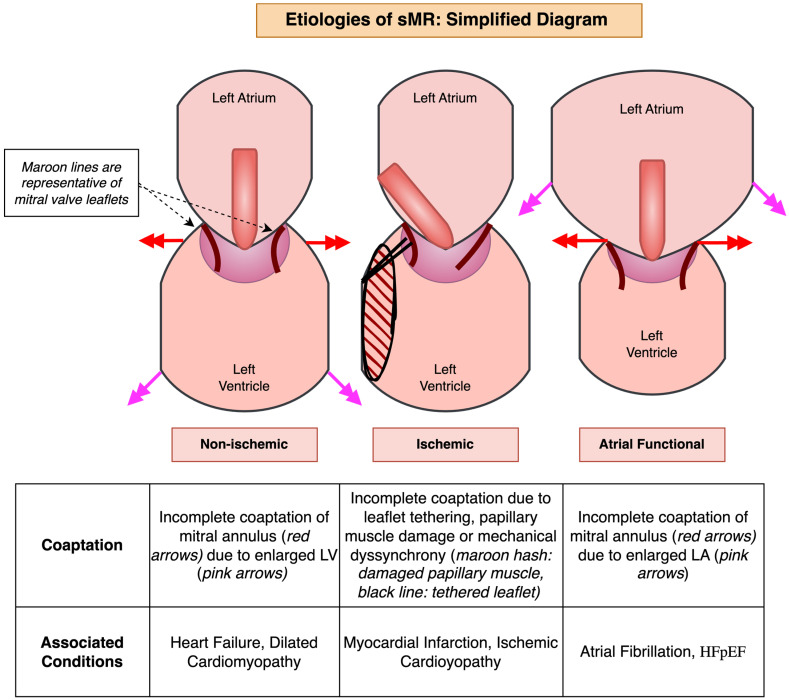
A simplified diagram demonstrating the three major etiologies of secondary mitral regurgitation (sMR).

**Figure 2 jcm-14-04518-f002:**
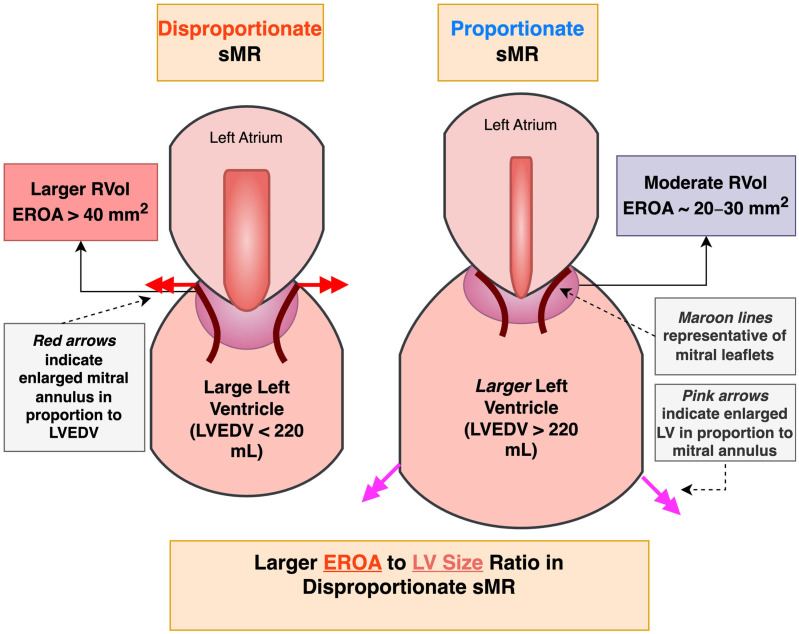
A simplified diagram demonstrating depicting proportionate versus disproportionate secondary mitral regurgitation (sMR). RVol—regurgitant volume, EROA—effective regurgitant orifice area, LVEDV—left ventricular end diastolic volume.

**Figure 3 jcm-14-04518-f003:**
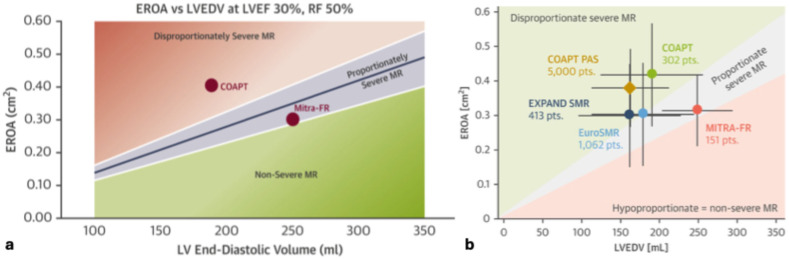
[Reprinted with permission of Grayburn et al. 2019 [69] and Stolz et al. 2025 [31]]. (**a**) shows the graph that was initially proposed by Grayburn et al. to illustrate the proportionate versus disproportionate theory for secondary MR. (**b**) includes the major RCTs and real-world registries, charted according to their EROA/LVEDV ratio representing the hypothetical proportionality. EROA: effective regurgitant orifice area, LVEDV: left ventricular end diastolic volume.

**Figure 4 jcm-14-04518-f004:**
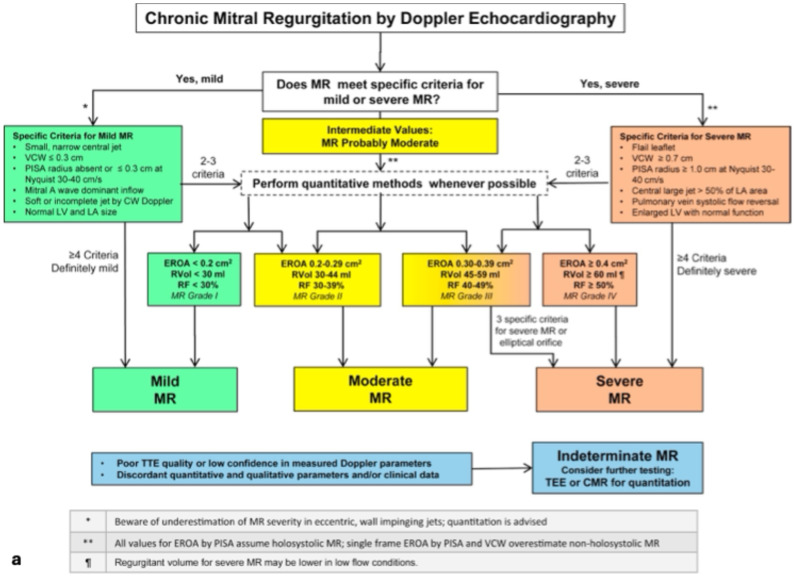
[Reprinted with permission from Zoghbi et al. 2017 [81]]—(**a**) is the 2017 ASE diagnostic algorithm for assessing MR severity presented side by side with (**b**) which is our proposed algorithm for diagnosing sMR in the context of m-TEER candidacy.

**Figure 5 jcm-14-04518-f005:**
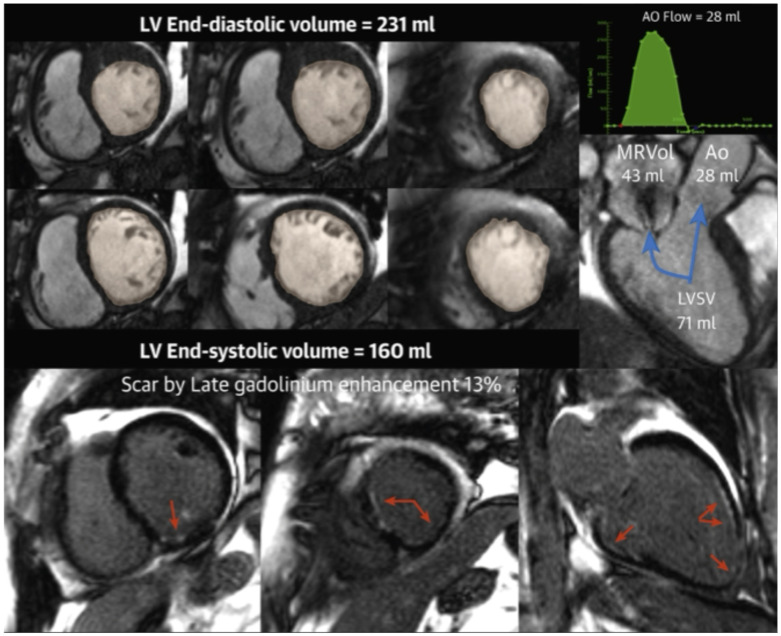
[Reprinted with permission from Tayal et al. 2020 [110]]: This example case outlines how cardiac magnetic resonance (CMR) can be used to quantitate secondary mitral regurgitation and scar. (**Top**) Left ventricular and total stroke volume (SV) quantification. (**Top right**) Phase contrast images are used to calculate the forward SV quantification. (**Bottom**) Late gadolinium enhancement was used to quantitate. This patient in this example was a 69-year-old woman with ischemic cardiomyopathy and secondary MR with RVol = 43 mL and RF = 52%. Scar burden = 13%. The patient expired 0.27 months after CMR. Blue arrows point towards the numeric values, which represent volumes quantitation. Red arrows represent scar tissue. Ao = aorta; LVSV = left ventricular stroke volume; MRVol = mitral regurgitant volume.

**Table 1 jcm-14-04518-t001:** Summary of major completed and ongoing trials assessing transcatheter edge-to-edge repair for the mitral valve (m-TEER) and novel device therapy in secondary MR (sMR).

M-TEER Trials					
Study Name (Years of Inclusion)	Design	Number of Patients	Comparison	Primary Endpoints	Conclusions
EVEREST II (2005–2008)	Multicenter RCT	279	M-TEER versus MV surgical repair or replacement in primary MR	Primary efficacy: composite of freedom from death, from surgery for MV dysfunction, and from grade 3+ or 4+ at 12 months; primary safety: composite of major adverse events within 30 days after procedure	M-TEER has a superior safety profile and comparable symptomatic improvement, but was less efficacious than MV surgery; M-TEER patients who were event free during the 1st year had more comparable efficacy outcomes to the MV surgery patients
TRAMI (2010–2013)	Prospective, Single Arm, Multicenter	722	Abbott MitraClip	Long-term mortality rates, cardiac rehospitalization, and reintervention	M-TEER demonstrated efficacy, improved functional status, and low re-intervention rates at 4 years; mortality was similar for ischemic and non-ischemic sMR
MITRA-FR (2013–2017)	Multicenter RCT	304	M-TEER + GDMT versus GDMT in sMR	Composite of death from any cause or unplanned HF hospitalization at 12 months	M-TEER did not demonstrate improvement in death or heart failure hospitalization
COAPT (2012–2017)	Multicenter RCT	614	M-TEER + GDMT versus GDMT in sMR	Primary efficacy endpoint: HF hospitalizations within 24 months; primary safety endpoint: freedom from device-related complications at 12 months	M-TEER demonstrated improvement in cardiovascular death, heart failure hospitalization, and symptoms at trial end and at 5 years
MITRA CRT (2015–2019)	Single Center RCT	31	M-TEER versus GDMT in ‘nonresponder’ patients with dilated cardiomyopathy + CRT + GDMT	Combination of CV death, heart transplantation, or HF hospitalization at 12 months	M-TEER demonstrated a reduction in the combined endpoint of CV death, heart transplantation, and HF hospitalizations
EURO SMR (2008–2020)	Multicenter, Retrospective registry	1628	M-TEER	MR severity, functional capacity, survival, and predictors of all-cause mortality	M-TEER is associated with long-term reduction in MR severity but the 5-year survival rate in this real-world registry was lower than that observed in the COAPT trial
COAPT-PAS (2019–2020)	Prospective, Single Arm, Multicenter	5000	Abbott MitraClip for sMR	Procedure success, MR severity, quality of life, HF hospitalization	High procedural success rates were demonstrated with COAPT-like and MITRA-FR like subgroups having lower hospitalizations at 1 year as compared to that of the COAPT and MITRA-FR RCTs, respectively
EXPAND (2018–2019)	Prospective, Single Arm, Multicenter	1041	Abbott MitraClip NTR or XTR Third Generation System	MR severity, functional capacity, quality of life, HF hospitalization, all-cause mortality	M-TEER demonstrated a greater degree of MR reduction as compared to EVERST II and COAPT
EXPAND G4 (2020–2022)	Prospective, Single Arm, Multicenter	1164	Abbott MitraClip Fourth Generation System	MR severity, functional capacity, quality of life, heart failure hospitalization, all-cause mortality	Fourth Generation M-TEER was demonstrated to be safe and effective with improvement in quality of life
MATTERHORN (2015–2022)	Multicenter RCT	210	M-TEER + GDMT versus surgical mitral valve repair or replacement + GDMT in sMR	Primary efficacy endpoint: composite of death, hospitalization for heart failure, MV reintervention, implantation of assist device, or stroke within 1 year after procedure; primary safety endpoint: composite of major adverse events within 30 days after the procedure	M-TEER is noninferior to mitral valve surgery in regard to efficacy, and had a superior safety profile within and after the periprocedural period
RESHAPE-HF2 (2017–2023)	Multicenter RCT	505	M-TEER + GDMT versus GDMT in sMR	Composite of first or recurrent hospitalization for heart failure or cardiovascular death during 24 months; rate of first or recurrent hospitalization for heart failure during 24 months; change from baseline to 12 months for quality of life	M-TEER demonstrated improvement in heart failure hospitalization and symptoms
CLASP IID (2018–2019)	Multicenter RCT	300	Abbot MitraClip versus Edwards PASCAL in primary MR	Primary efficacy: MR severity at 6 months; primary safety: adverse events within 30 days	Edwards PASCAL demonstrated noninferiority to Abbott MitraClip
miCLASP (2019–2024)	Prospective, Single Arm, Multicenter	544	Edwards PASCAL	Primary efficacy endpoint: MR severity; primary safety endpoint: major adverse event within 30 days	M-TEER demonstrated safety and efficacy with reduction in MR, hospitalization, and symptoms
**Other Percutaneous Treatments for sMR**					
Transcatheter mitral loop cerclage (2015–2016)	Prospective, Single Arm, Single Center	5	Mitral Loop Cerclage Annuloplasty for sMR	MR severity and chamber dimensions over 6 months	Procedural success in 4 out of 5 patients with reverse remodeling and electrical remodeling
Cardioband (2013–2016)	Prospective, Single Arm, Multicenter	60	Edwards Lifesciences Cardioband for sMR	MR severity, survival, and functional capacity at 1 year	Cardioband demonstrated improvement in MR and functional capacity
Intrepid TMVR (2015–2017)	Prospective Single Arm, Multicenter	50	Transcatheter Mitral Valve Replacement for sMR	Procedural success, MR severity, and symptoms improvement	TMVR is feasible in high or extreme risk patients; APOLLO RCT comparing TMVR to m-TEER is ongoing
MAVERIC (2013–2017)	Prospective, Single Arm, Multicenter	45	ARTO System (transcatheter annular reduction device) for sMR	Primary efficacy: MR reduction at 30 days; primary safety: adverse event within 30 days after procedure	ARTO System is safe and had sustained efficacy with reduction in MR, LVEDV, and heart failure hospitalizations at 2 years
REDUCE-FMR (2014–2018)	Multicenter RCT	120	Carillion Mitral Contour System (coronary-sinus-based percutaneous annuloplasty) +GDMT vs. GDMT for sMR	MR severity at 12 months	The Carrilion device significantly reduced sMR as compared to GDMT

**Table 2 jcm-14-04518-t002:** Comparison of major randomized control trials (RCTs) for transcatheter edge-to-edge repair for the mitral valve (m-TEER) with information regarding inclusion/exclusion criteria, clinical characteristics, and outcomes. EROA: effective regurgitant orifice area, RVol: regurgitant volume, NYHA: New York Heart Association, ACEI: angiotensin-converting enzyme inhibitor, ARB: angiotensin II receptor blocker, ARNI: angiotensin receptor/neprilysin inhibitor, MRA: mineralocorticoid receptor antagonist, SGLT-2: sodium glucose transport 2, ICD: implanted cardioverter-defibrillator, CRT: cardiac resynchronization therapy, PCI: percutaneous coronary intervention, CABG: coronary artery bypass grafting, eGFR: estimated glomerular filtration rate, KCCQ: Kansas City Cardiomyopathy Questionnaire.

Comparison of Trial Design of Major m-TEER RCTs			
Inclusion Criteria	COAPT	MITRA-FR	RESHAPE-HF2
MR Severity	At least moderate to severe	At least moderate to severe	MR Severity
Definition for moderate to severe (3+) MR	EROA > 30 mm^2^ and/or RVol > 45 mL (As per ACC/AHA 2006/2008 Guideline)	EROA > 20 mm^2^ and/or RVol > 30 mL (As per ESC/EACT 2012 Guidelines)	EROA > 30 mm^2^ and/or RVol > 45 mL (As per EACVI 2012 Guidelines)
NYHA Stage	II, III, IVa (ambulatory)	II, III, or IV	II, III, IV
LV EF (%)	15–40	20–50	15–35 if NYHA II, 15–45 if NYHA III or IV
GDMT at baseline	Uptitrated to maximum dose, stable dosage, CRT if eligible	Variable GDMT adjustment, ‘real world’ practice	Uptitrated to maximum dose, stable dosage, CRT if eligible
**Exclusion Criteria**			
LV end systolic dimension (mm)	>70 mm	-	-
**Primary Endpoint(s)**	Primary efficacy: HF hospitalization within 24 months	Composite rate of all-cause mortality or HF hospitalization at 12 months	Composite rate of recurrent HF hospitalizations and cardiovascular death within 24 months
	Primary safety: Freedom from device related complications at 12 months	-	HF hospitalizations within 24 months
	-	-	Change in KCCQ score at 12 months
**Comparison of Clinical Characteristics**			
**Clinical Characteristics (Average)**	COAPT (*n* = 614)	MITRA-FR (*n* = 304)	RESHAPE-HF2 (*n* = 505)
ACEI, ARB, or ARNI (% of total)	67%	-	82%
ACEI or ARB	-	74%	74%
Beta Blocker (% of total)	90%	90%	96%
Diuretics (% of total)	89%	99%	95%
MRA (% of total)	50.2%	54.6%	82.4%
SGLT-2 Inhibitor (% of total)	-	-	9%
Previous ICD (% of total)	31.3%	34.5%	35.2%
Previous CRT (% of total)	36.5%	26.6%	29.1%
Previous PCI (% of total)	48.2%	46.1%	44.4%
Previous CABG (% of total)	26.3%	40.2%	-
EROA (mm^2^)	41 ± 15	31 ± 15	25
LV end-diastolic volume (mL)	194	252	211
LVEF (%)	31 ± 9	33 ± 6	31 ± 8
eGFR (mL/min/1.73 m^2^)	49 ± 26	50 ± 20	56 ± 21
NT-proBNP (pg/mL)	5174 ± 6567	3407 (1948–6790)	4185 ± 4340
Proportion of patients with severe MR, EROA ≥ 40 mm^2^ (% of total)	41	16	14
Non-ischemic cause of cardiomyopathy (%)	39.3	59.2	50.6
**Outcomes**			
**Outcome**	COAPT	MITRA-FR	RESHAPE-HF2
Primary Efficacy	0.3 (0.40–0.70) *p* < 0.001	1.16 (0.69 –1.84) *p* = 0.53	0.64 (0.48–0.85) *p* = 0.002 ***
Primary Safety	96.6% freedom vs. performance goal of 88%, *p* < 0.001	- **	- ****
All-cause Mortality	0.62 (0.46–0.82) *p* < 0.001 0.72 (0.58–0.89) at 5 years*	1.11 (0.69–1.77)	0.90 (0.71–1.13) [per 100 patient years]
Cardiovascular Mortality	0.59 (0.43–0.81) *p* = 0.001 0.71 (0.56–0.90) at 5 years *	1.09 (0.67–1.78)	0.84 (0.55–1.28) *p* = 0.43
Any-cause Hospitalization	0.77 (0.64–0.93) *p* = 0.01 0.75 (0.63–0.89) at 5 years *	-	0.82 (0.63–1.07) *p* = 0.15 [per 100 patient years]
HF Hospitalization	0.53 (0.40–0.67) *p* < 0.001 0.49 (0.40–0.61) at 5 years *	1.13 (0.81–1.56)	0.59 (0.42–0.82) *p* = 0.002 [per 100 patient years]
Change in KCCQ score at 12 months	16.1 (11.0–21.2) *p* < 0.001	No significant difference on EQ5D Score	10.9 (6.8–15.0) *p* < 0.001
Change in 6 Minute Walk Distance from Baseline to 12 months (meters)	57.9 (32.7–83.1) *p* < 0.001	No significant difference	20.5 (0.3–40.7) *p* = 0.05
MV Surgery at 24 months	0.14 (0.02–1.17) *p* = 0.57	-	0.51 (0.05–5.58) *p* = 0.57

* Data from the 5-year follow-up of COAPT study. ** There was no primary safety efficacy endpoint in MITRA-FR, but periprocedural complications occurred in 14.6% of patients in the intervention group. *** In the ‘Primary Efficacy’ row for RESHAPE-HF2, only the primary efficacy endpoint ‘Rate of the composite of heart failure hospitalization or death from cardiovascular causes’ is listed, the other two primary efficacy endpoints are in the following columns. **** There was no primary safety efficacy endpoint in RESHAPE-HF2, but periprocedural complications occurred in 1.6% of patients in the intervention group.

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
