# Peer review of "Diagnosis and Diagnostic Challenges of Secondary Mitral Regurgitation in the Era of Transcatheter Edge-to-Edge Repair of the Mitral Valve"

_jcm, 2025, doi:10.3390/jcm14134518_

Round 1

Reviewer 1 Report

Comments and Suggestions for Authors

Overall Impression:

This represents a very strong and timely review article. The expanded discussion on diagnostic challenges, particularly your proposed algorithm and the detailed exploration of CMR and AI, significantly enhances the manuscript's value and sorts the whole topic into the modern times. The authors have clearly put considerable thought into navigating the complexities of sMR diagnosis in the current era.

Strengths of the review can be summarise as follows:

  • Proposed Diagnostic Algorithm (Figure 4b): Developing a specific and "easy to follow" algorithm for sMR that accounts for the nuances of heart failure patients is highly valuable and addresses a significant clinical need. It demonstrates a clear understanding of the limitations of existing guidelines in this specific context.
  • Detailed Discussion of Echocardiographic Limitations: Your critical assessment of various echocardiographic parameters (LA/LV size, mitral inflow velocity, eccentric jets, PVSFR, PISA) in the context of sMR is excellent and highlights the challenges faced by clinicians.
  • Emphasis on Quantitative Methods: The strong recommendation for quantitative volumetric methods (RVol, RF) and the explanation of why they are often superior to PISA in sMR is well-justified and important.
  • In-Depth Review of CMR: The dedicated section on CMR as the "gold standard" for sMR quantification and its role in assessing myocardial scar and prognostication is very comprehensive and a major asset to the article. Figure 5 effectively illustrates this point.
  • Forward-Looking AI Section: The inclusion of the emerging role of artificial intelligence in sMR phenotyping and risk stratification is highly relevant and demonstrates foresight, positioning the review at the cutting edge.

I have some minor comments for improvement: 

  • Please adjust figure 1 and 2. I believe that the graphical display of LA and LV, as well as leaflets are not clear. The differences in LA size between vSMR and aSMR is not clear and needs to be accentuated. Besides, it would be good to add leaflets to demonstrate the differences in teathering/tenting etc.. Consider adding very brief labels within the diagram to explicitly link the visual representation to the mechanisms of sMR. This would make it even more intuitive for a quick visual reference.
  • Please add a short paragraph on heart failure medication and emphasize, that a detailed imaging assessment of MR severity should be performed after OMT.
  • Maybe consider adding a small paragraph on multivalvular disease in sMR patients. Some of these patients have AS and there is data showing, that addressing AS first can lead to reduction of sMR severity.
  • Figure 4b: This represents one strength of your manuscript. As "b" this important figure is easily overread. Consider presenting it as a seperate figure.
  • 3D echo: You briefly touch upon 3D echo at line 317. Given its increasing role in accurate quantification of MR orifice area and volume, consider slightly expanding on its specific advantages in sMR, even with its limitations. For instance, it can overcome some of the geometric assumptions of 2D PISA.
  • please expand on your conclusion part: Stress the increasing value of multimodality imaging in heart failure with sMR. Please also consider underlining the complexity of accurately assessing severity of sMR and necessity of an intervention in echo highlighting the potential need for an early referral to a high volume center for additional MR assessment.

Author Response

Please see document in zip folder below: Reviewer Suggestions and Author Response

Reviewer 2 Report

Comments and Suggestions for Authors

In this interesting review, the authors exhaustively described the etiology, management, and diagnosis of sMR in the era m-TEER.

They described the main trials that evaluated the prognostic role of m-TEER, particularly focusing on COAPT, MITRA-FR and RESHAPE-HF2.

Interestingly, COAPT demonstrated cardiovascular mortality and HF hospitalization benefit with M-TEER, whereas RE-SHAPE-HF2 only found HF hospitalization benefit.

MITRA-FR’s study population had comparatively more advanced cardiomyopathy and worse medical optimization.

A key message of the authors is the following one: The disproportionate sMR to LV size or indirectly to the existing cardiomyopathy (larger EROA and lower LV size) is more amenable to MV intervention, whereas sMR proportionate to LV dilation (lower EROA and higher LV size) is more amenable with pharmacologic therapy. MITRA-FR showed no benefit for m-TEER in patients with a lower EROA.

EROA >30 mm2 predicts a higher likelihood of response to therapy.

The manuscript is well-structured, the tables and figures are clear, the references are appropriate and the conclusions are supported by the large number of information provided in the review.

I have only one suggestion for the authors.

At the end of the paragraph 4.1. "Defining Secondary MR", on line 319, the authors could also mention the potential usefulness of the innovative modified Haller index (MHI) for identifying, among the individuals with mitral valvulopathy, those with lower incidence of adverse cardiovascular events over mid-to-long term follow-up, particularly those with a narrow antero-posterior (A-P) thoracic diameter. A simple measurement of the A-P thoracic diameter from the echocardiographic parasternal long-axis view may allow the cardiologists to indentify individuals with lower EROA and lower LV size, generally associated with good outcome (PMID: 35669134). Noninvasive chest shape assessment should be considered for implementation in clinical practice. It is important to consider not only intrinsic determinants of mitral valve disease but also extrinsic determinants of mitral valve disease. It is likely that, also among sMR patients, those with a concave-shaped chest wall conformation and/or various degrees of anterior wall deformity might be affected by less severe forms of mitral valve disease. Further studies are needed for evaluating the effect of chest shape conformation (in particular a narrow A-P thoracic diameter) on sMR degree and prognosis.

Author Response

(The authors gave the same response as above.)
